# VALL-E 2: Neural Codec Language Models are Human Parity Zero-Shot Text to Speech Synthesizers

## Abstract

This paper introduces VALL-E 2, the latest advancement in neural codec language models that marks a milestone in zero-shot text-to-speech synthesis (TTS), *achieving human parity for the first time*. Based on its predecessor, VALL-E, this work introduces two significant enhancements: **Repetition Aware Sampling** refines the original nucleus sampling process by accounting for token repetition in the decoding history. It not only stabilizes the decoding but also circumvents the infinite loop issue. **Grouped Code Modeling** organizes codec codes into groups to effectively shorten the sequence length, which not only boosts inference speed but also addresses the challenges of long sequence modeling. Our experiments on the LibriSpeech and VCTK datasets show that VALL-E 2 surpasses previous systems in speech robustness, naturalness, and speaker similarity. It is the first of its kind to reach human parity on these benchmarks. Moreover, VALL-E 2 consistently synthesizes high-quality speech, even for sentences that are traditionally challenging due to their complexity or repetitive phrases. The advantages of this work could contribute to valuable endeavors, such as generating speech for individuals with aphasia or people with amyotrophic lateral sclerosis. See `https://anonymous/valle2` for demos of VALL-E 2.

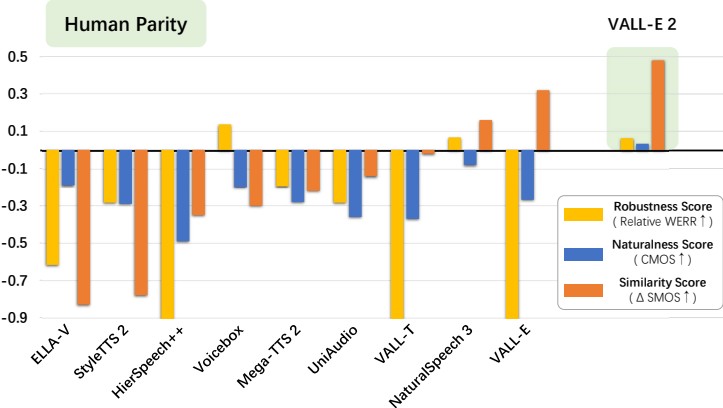

Figure 1: VALL-E 2 achieves human parity zero-shot TTS performance for the first time. Robustness, naturalness and similarity scores are relative numbers calculated based on the results reported in the original papers, irrespective of differences in model architecture and training data, such as $\triangle$ SMOS(ELLA-V) = SMOS(ELLA-V) $-$ SMOS(GroundTruth).

## 1 Introduction

Text-to-speech synthesis (TTS) aims to generate high-quality speech from text input with a high degree of clarity and intelligibility. Along with the progress of deep learning, significant improvements have been made in TTS research in recent years (Shen et al., 2018; Li et al., 2019; Ren et al., 2019). Some systems, trained with clean single-speaker speech data recorded in sound-recording studios, have even achieved human-level quality for single-speaker speech generation (Tan et al., 2024). However,

zero-shot TTS, which requires the model to synthesize speech for unseen speakers using a short enrolled speech sample during inference, remains a challenging problem.

The previous work, VALL-E (Wang et al., 2023a), marked a significant breakthrough in this area. It is capable of synthesizing personalized speech using only a 3-second recording, while preserving the speaker's voice, emotion, and acoustic environment. VALL-E is a neural codec language model that represents speech signals as discrete codec codes with a neural audio codec model. Specifically, it trains an autoregressive language model to generate the coarse codec codes and another non-autoregressive model to generate the remaining fine codec codes. Instead of using greedy search, which continually generates silence codec codes, VALL-E uses random sampling for model inference. However, VALL-E has two key limitations: 1) Stability: The random sampling used during inference can lead to instability in output, while nucleus sampling with a small top-p value may cause an infinite loop issue. This can be mitigated by multiple-time sampling and subsequent sorting, but this approach increases the computational cost. 2) Efficiency: The autoregressive architecture of VALL-E is bound to the same high frame rate as the off-the-shelf audio codec model, which cannot be adjusted, resulting in a slower inference speed.

Several follow-up works have been proposed to address these problems (Song et al., 2024; Xin et al., 2024; Borsos et al., 2023; Le et al., 2024; Ju et al., 2024). To improve stability, some works leverage text-speech alignment information in model training and inference (Song et al., 2024; Xin et al., 2024). These methods, relying on a forced-alignment model, inevitably introduces errors in the alignment result, which could affect the final performance. It also complicates the overall architecture and increases the burden for data scaling up. To improve modeling efficiency, some works explore fully non-autoregressive methods for zero-shot TTS (Borsos et al., 2023; Le et al., 2024; Ju et al., 2024). However, these methods require frame-aligned text-speech data for model training, facing the same problem as discussed before. Additionally, the non-autoregressive model generates the tokens with a pre-determined duration result, which constrains the search space of the generated speech and sacrifices the prosody and naturalness.

In this work, we propose VALL-E 2, the first human parity zero-shot text-to-speech synthesis system. Building upon its predecessor VALL-E, VALL-E 2 employs a neural codec language modeling method for speech synthesis and incorporates two key modifications: repetition aware sampling and grouped code modeling. Repetition aware sampling, an improvement over the random sampling used in VALL-E, adaptively employs either random or nucleus sampling for each time step token prediction. This selection is based on the token repetition in the decoding history, enhancing the stability of the decoding process and circumventing the infinite loop issue encountered in VALL-E. Grouped code modeling, on the other hand, partitions the codec codes into groups, each of which is modeled in a single frame in the AR modeling process. This approach not only accelerates inference by reducing the sequence length but also improves performance by mitigating the long context modeling problem. Notably, VALL-E 2 requires only simple utterance-wise speech-transcription pair data for training, greatly simplifying the process of collecting and processing training data and facilitating potential scalability.

VALL-E 2 is trained on the large-scale Libriheavy dataset (Kang et al., 2024). Subsequent evaluations demonstrate that it achieves performance on par with human capabilities on both the in-domain LibriSpeech dataset (Panayotov et al., 2015) and the out-of-domain VCTK datasets (Veaux et al., 2016). As illustrated in Figure 1, VALL-E 2 significantly outperforms VALL-E and other prior works on the LibriSpeech dataset in terms of robustness, naturalness, and similarity score, even achieving human parity performance. The numbers in Figure 1 are relative numbers ($\triangle$ Score(Model) = Score(Model) − Score(GroundTruth)) based on the results reported in the paper. In this context, human parity indicates that the robustness, naturalness, and similarity metrics of VALL-E 2 surpass those of the ground truth samples (meaning that $\triangle$ WERR(VALL-E 2) > 0, $\triangle$ CMOS(VALL-E 2) > 0, and $\triangle$ SMOS(VALL-E 2) > 0), meaning that VALL-E 2 can generate accurate, natural speech in the exact voice of the original speaker, comparable to human performance. It is important to note that this conclusion is drawn solely from experimental results on the LibriSpeech and VCTK datasets. Moreover, VALL-E 2 can accelerate the decoding process by multiple times with almost no performance degradation. To specifically evaluate the stability of VALL-E 2, we synthesize speech for complex sentences that are hard to read or contain many repeated phrases, and found that VALL-E 2 can always stably generate high-quality speech. The benefits of this work could support meaningful initiatives, such as generating speech for individuals with aphasia or people with amyotrophic lateral sclerosis.

## 2 RELATED WORK

### 2.1 ZERO-SHOT TTS

Early work in zero-shot TTS typically employed speaker adaptation and speaker encoding methods, which often required additional fine-tuning, complex pre-designed features, or heavy structure engineering (Chen et al., 2019; Wang et al., 2020; Arik et al., 2018; Casanova et al., 2022). Inspired by the success of Large Language Models (LLMs) in natural language processing, VALL-E (Wang et al., 2023a; Zhang et al., 2023b) represented speech as discrete codec codes with an off-the-shelf neural codec model, and approached TTS as a conditional codec language modeling task. This approach allowed VALL-E to train a codec language model on large-scale training data and perform zero-shot TTS via prompting, achieving significant zero-shot TTS capability.

This breakthrough inspired subsequent research works to address zero-shot TTS through a language modeling approach. For instance, VALL-E X (Zhang et al., 2023b) extended VALL-E to cross-lingual TTS tasks with an additional language ID token. SPEAR-TTS (Kharitonov et al., 2023) and Make-a-voice (Huang et al., 2023a) leveraged semantic units from a speech self-supervised model as an intermediate interface between text and acoustic codec codes, enabling better training data efficiency. Mega-TTS (Jiang et al., 2023b) and Mega-TTS 2 (Jiang et al., 2023a) proposed to first disentangle the multiple attributes in speech, then only model partial attributes with a language modeling approach. ELLA-V (Song et al., 2024) and RALL-E (Xin et al., 2024) improved VALL-E's robustness and stability by including speech-text alignment prediction into the decoding process. UniAudio (Yang et al., 2023b) and BASE TTS (Łajszczak et al., 2024) further explored scaling the codec language model to 1b parameters and 100k hours of training data.

Meanwhile, other works explored fully non-autoregressive modeling methods to accelerate the inference speed. For example, Soundstorm (Borsos et al., 2023) leveraged the confidence-based parallel decoding scheme (Chang et al., 2022) to generate the acoustic codec codes with a non-autoregressive model. StyleTTS 2 (Li et al., 2024), UniCATS (Du et al., 2024a), NaturalSpeech 2 (Shen et al., 2023) and NaturalSpeech 3 (Ju et al., 2024) used diffusion model (Ho et al., 2020) for the prompt-conditioned text to speech synthesis. Voicebox (Le et al., 2024) and Audiobox (Vyas et al., 2023) used flow-matching method (Lipman et al., 2022) and achieved better speech modeling capability. In this work, VALL-E 2 follows the codec language modeling method of VALL-E, and enables a stable decoding process without the need for complex speech data processing and preparation, such as duration or pitch information used in previous methods. Notably, VALL-E 2 is the first to successfully achieve human parity in zero-shot TTS on both LibriSpeech and VCTK datasets.

### 2.2 CODEC-BASED SPEECH MODELS

Inspired by the promising performance of neural codec codes in zero-shot TTS, many subsequent research works have started to explore its effectiveness on more speech tasks. For instance, PolyVoice (Dong et al., 2023) adopted VALL-E and built a codec-based language model for speech-to-speech translation. SpeechX (Wang et al., 2023c) extended VALL-E with multi-task learning, demonstrating efficacy in zero-shot TTS, noise suppression, target speaker extraction, speech removal, and speech editing tasks. In addition to speech generation, VioLA (Wang et al., 2023b) further explored codec-based speech models for speech understanding tasks, unifying codec language models for speech recognition, synthesis, and translation tasks. AudioPaLM (Rubenstein et al., 2023) fused the codec tokens into the LLM PaLM 2 (Anil et al., 2023), and demonstrated promising results on speech recognition and translation tasks.

These works typically employ SoundStream (Zeghidour et al., 2021) and Encodec (Défossez et al., 2022), initially designed for speech compression, as the neural codec model. Inspired by these successes, several works have proposed more novel neural codecs specifically for speech processing tasks. These include Vocos (Siuzdak, 2023), SpeechTokenizer (Zhang et al., 2023a), AudioDec (Wu et al., 2023), AcademiCodec (Yang et al., 2023a), Descript-audio-codec (DAC) (Kumar et al., 2024), FunCodec (Du et al., 2024b), and RepCodec (Huang et al., 2023b). The Codec-SUPERB challenge (Wu et al., 2024) was announced to benchmark various codec codes across a wide range of speech tasks. In this work, we utilize the Encodec model to tokenize speech signals and the Vocos decoder to generate target high-quality speech signals.

## 3 VALL-E 2

### 3.1 Problem Formulation: Grouped Codec Language Modeling

Following VALL-E, we use an off-the-shelf neural audio codec model to represent speech signals as discrete codec code sequence, and regard TTS as a conditional codec language modeling task. To improve the efficiency, VALL-E 2 introduce a grouped codec language modeling method, where we partition the codec code sequence into groups of a certain size, and model each group of codec codes as one frame. In this way, we can get rid of the frame rate constraint of the off-the-shelf neural audio codec model, and reduce the frame rate by integer multiples. It is not only beneficial for the inference efficiency but also the overall speech quality by mitigating the long context modeling problem.

With TTS training objective, VALL-E 2 is optimized to maximize the likelihood of the grouped code sequence given the text condition. Specifically, given an audio sample $\mathbf{y}$ and its corresponding tokenized text transcription $\mathbf{x} = [x_0, x_1, \ldots, x_{(L-1)}]$, where $L$ is the text sequence length, we first use a pre-trained neural audio codec model to convert the audio sample $\mathbf{y}$ into a codec code sequence $\mathbf{C}^{T \times J} = [\mathbf{c}_0, \mathbf{c}_1, \ldots, \mathbf{c}_{(T-1)}]$, where $T$ is the code sequence length, $J$ (here $J = 8$) is the number of the quantizers in the codec model, and each $\mathbf{c}_t$ represents the 8 codes for each time step. Then we partition it into the grouped code sequence $\mathbf{C}^G = [\mathbf{C}_{0:G}, \mathbf{C}_{G:2G}, \ldots, \mathbf{C}_{(T-G):T}]$ with the group size $G$, and $\mathbf{C}_{0:G}$ stands for the group $[\mathbf{c}_0, \mathbf{c}_1, \ldots, \mathbf{c}_{(G-1)}]$. Due to the typical short silence at the start of an utterance, we can clip a few codes from the start of the code sequence to let the code sequence length $T$ be the integer multiple of the group size without removing any speech information. Finally, we train the VALL-E 2 model $\theta$ to minimize the negative log-likelihood of the grouped code sequence $\mathbf{C}^G$ conditioned on the text sequence $\mathbf{x}$:

$$\mathcal{L} = -\log p(\mathbf{C}^G | \mathbf{x}; \theta) \tag{1}$$

$$= -\sum_{t=0}^{T/G-1} \log p(\mathbf{C}_{t \cdot G:(t+1) \cdot G} | \mathbf{C}_{<t \cdot G}, \mathbf{x}; \theta), \tag{2}$$

where $\mathbf{C}_{t \cdot G:(t+1) \cdot G}$ is the $t$-th group of codec codes $[\mathbf{c}_{t \cdot G}, \ldots, \mathbf{c}_{((t+1) \cdot G-1)}]$, and $\mathbf{C}_{<t \cdot G}$ is all the codec codes in the previous $(t-1)$ groups.

During inference, VALL-E 2 performs zero-shot TTS task via prompting. Given a text input (containing both the transcription of speech prompt and the text to synthesis) and grouped codec codes from an unseen speaker, serving as the condition and prompt, the model can generate the target grouped codec codes with the corresponding content and speaker's voice. Specifically, given the text sequence $\mathbf{x}$ and the enrolled speech sample of the unseen speaker $\mathbf{y}'$, we can obtain the corresponding grouped code sequence $\mathbf{C}^P = \mathbf{C}^G_{<T'} = [\mathbf{C}_{0:G}, \mathbf{C}_{G:2G}, \ldots, \mathbf{C}_{(T'-G):T'}]$. Then, We generate the target grouped code sequence $\mathbf{C}^T = \mathbf{C}^G_{\geq T'} = [\mathbf{C}_{T':(T'+G)}, \ldots, \mathbf{C}_{(T-G):T}]$ conditioned on the text sequence $\mathbf{x}$ and code prompt $\mathbf{C}^P$:

$$\mathbf{C}^T = \arg\max_{\mathbf{C}} p(\mathbf{C} | \mathbf{C}^P, \mathbf{x}; \theta) \tag{3}$$

$$= \arg\max_{\mathbf{C}} \sum_{t=T'/G}^{T/G-1} \log p(\mathbf{C}_{t \cdot G:(t+1) \cdot G} | \mathbf{C}_{<t \cdot G}, \mathbf{x}; \theta). \tag{4}$$

Finally, we can convert the target code sequence $\mathbf{C}^T$ to the target speech waveform using an off-the-shelf neural codec decoder.

### 3.2 VALL-E 2 Architecture

Building upon VALL-E, VALL-E 2 also use a hierarchical structure: an Autoregressive (AR) codec language model and a Non-Autoregressive (NAR) codec language model. The AR model generates sequence of the first codec code for each frame in an autoregressive manner, while the NAR model generates each remaining code sequence based on the preceding code sequences in a non-autoregressive manner. Both models utilize the same Transformer architecture with a text embedding layer, a code embedding layer, and a code prediction layer. We use distinct embeddings for the codes from different codec quantizers and share the parameters of the code prediction layer

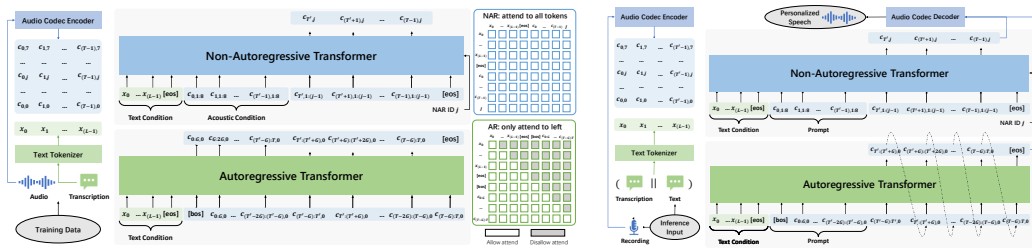

(a) Training overview.        (b) Inference overview.

Figure 2: Overview of VALL-E 2, consisting of an autoregressive and a non-autoregressive Transformer. The autoregressive Transformer is designed to generate grouped codec codes. The repetition aware sampling method is proposed to predict grouped code sequence during autoregressive model inference.

with the parameters of the code embedding layer. In addition, the AR model has a group embedding layer to project the code embedding to the group embedding, and a group prediction layer for the prediction of codes in one group . The NAR model has a code ID embedding layer to specify the ID of the code sequence to predict. The AR model and NAR model have different attention mask strategies: the AR model uses the causal attention strategy and the NAR model uses the full attention strategy, as shown in the right part of Figure 2a.

### 3.3 VALL-E 2 TRAINING

Figure 2a shows the overview of VALL-E 2 model training. It is noteworthy that the training of VALL-E 2 requires only simple utterance-wise speech-transcription pair data, without any complex data such as force-alignment result or additional audio clips of the same speaker for reference. This greatly simplifies the process of collecting and processing training data.

Specifically, for each audio and corresponding transcription in the training dataset, we initially utilize the audio codec encoder and text tokenizer to obtain the codec codes $\mathbf{C} = [\mathbf{c}_0, \mathbf{c}_1, \ldots, \mathbf{c}_{(T-1)}]$ and the text sequence $\mathbf{x} = [x_0, x_1, \ldots, x_{(L-1)}]$, respectively. These are then used for the AR model and the NAR model training.

#### 3.3.1 AUTOREGRESSIVE MODEL TRAINING

The AR model is trained to predict the first codec code sequence $\mathbf{c}_{:,0} = [c_{0,0}, c_{1,0}, \ldots, c_{(T-1),0}]$ conditioned on the text sequence $\mathbf{x}$ in an autoregressive manner.

As shown in the lower middle part of Figure 2a, we first obtain the text embedding sequence $\mathbf{E}^x = [\mathbf{e}_0^x, \mathbf{e}_1^x, \ldots, \mathbf{e}_{(L-1)}^x]$ and the code embedding sequence $\mathbf{E}^c = [\mathbf{e}_0^c, \mathbf{e}_1^c, \ldots, \mathbf{e}_{(T-1)}^c]$ using the text embedding matrix $\mathbf{W}^x$ and the code embedding matrix $\mathbf{W}^c$.

$$\mathbf{e}_l^x = \mathbf{W}^x \odot x_l, \tag{5}$$
$$\mathbf{e}_t^c = \mathbf{W}^c \odot c_{t,0}, \tag{6}$$

where $l$ and $t$ denotes the indices of each item in the text sequence and code sequence, respectively, and $\odot$ denotes index selection. Then, we partition the code embedding sequence into groups of size $G$, concatenate each group of the the code embeddings in the hidden dimension, and obtain the group embedding sequence $\mathbf{E}^g = [\mathbf{e}_0^g, \mathbf{e}_1^g, \ldots, \mathbf{e}_{(T/G-1)}^g]$ using the group embedding matrix $\mathbf{W}^g$.

$$\mathbf{e}_t^g = \mathbf{e}_{t \cdot G:(t+1) \cdot G}^c \cdot \mathbf{W}^g. \tag{7}$$

We concatenate the text embedding sequence $\mathbf{E}^x$ and the group embedding sequence $\mathbf{E}^g$, inserting the embedding of special tokens $<eos>$ and $<bos>$ in between:

$$\mathbf{E}^0 = \mathbf{E}^x \parallel [\mathbf{e}_{<eos>}, \mathbf{e}_{<bos>}] \parallel \mathbf{E}^g, \tag{8}$$

where $\parallel$ indicates concatenation in the temporal dimension. We then separately add the learnable position embedding to the text embedding sequence and the group embedding sequence. The AR model is fed with $\mathbf{E}^0$ and trained to predict corresponding code sequence with a special token $<eos>$

appended at the end using a linear mapping group prediction layer and softmax code prediction layer. Due to the causal attention mask strategy, the prediction of each code group $\mathbf{c}_{t \cdot G:(t+1) \cdot G, 0}$ can only attend to the text sequence $\mathbf{x}$ and the preceding codes $\mathbf{c}_{<t \cdot G, 0}$, as demonstrated in the lower right part of Figure 2a.

Overall, the parameters $\theta_{\mathrm{AR}}$ of the AR model is optimized by minimizing the negative log likelihood of the first code sequence $\mathbf{c}_{:,0}$ conditioned on the text sequence $\mathbf{x}$:

$$\mathcal{L}_{\mathrm{AR}} = -\log p(\mathbf{c}_{:,0}|\mathbf{x}; \theta_{\mathrm{AR}}) \tag{9}$$

$$= -\sum_{t=0}^{T/G-1} \sum_{t'=t \cdot G}^{(t+1) \cdot G - 1} \log p(c_{t',0}|\mathbf{c}_{<t \cdot G, 0}, \mathbf{x}; \theta_{\mathrm{AR}}). \tag{10}$$

In the AR model of VALL-E 2, the group sequence $\mathbf{c}_{:,0} = [\mathbf{c}_{0:G}, \mathbf{c}_{G:2G,0}, \ldots, \mathbf{c}_{(T-G):T,0}]$ is modeled in an autoregressive approach, while the codec codes within each group $\mathbf{c}_{t \cdot G:(t+1) \cdot G, 0} = [c_{t \cdot G, 0}, c_{(t \cdot G+1), 0} \ldots, c_{((t+1) \cdot G - 1), 0}]$ are modeled in a non-autoregressive way.

### 3.3.2 NON-AUTOREGRESSIVE MODEL TRAINING

Given the first code sequence generated by the AR model, the NAR model is trained to generate remaining code sequence $\mathbf{c}_{:,j}$ for each codec code ID $j$ conditioned on the text sequence $\mathbf{x}$ and the preceding code sequences $\mathbf{c}_{:,<j}$ in a non-autoregressive manner, where $j \in [1, \ldots, 7]$.

As we have access to all 8 code sequences of the prompt during inference, to better model the speaker information of the prompt, during training, we explicitly split all the code sequences $\mathbf{C}$ into an acoustic condition $\mathbf{C}_{<T'}$ and target code sequences $\mathbf{C}_{\geq T'}$ with a randomly sampled length $T'$. The model is then optimized to predict each target code sequence $\mathbf{c}_{\geq T', j}$ conditioned on the text sequence $\mathbf{x}$, all $J = 8$ code sequences in the acoustic condition $\mathbf{C}_{<T'}$ and the preceding target code sequences $\mathbf{C}_{\geq T', <j}$ in a non-autoregressive manner.

As shown in the upper middle part of Figure 2a, we first obtain the text embedding sequence $\mathbf{E}^x = [\mathbf{e}_0^x, \mathbf{e}_1^x, \ldots, \mathbf{e}_{(L-1)}^x]$ using the text embedding matrix $\mathbf{W}^x$, as denoted in Equation 5. Then, we obtain the code embedding sequence $\mathbf{E}^c = [\mathbf{e}_0^c, \mathbf{e}_1^c, \ldots, \mathbf{e}_{(T-1)}^c]$ by obtaining all the code embeddings in the acoustic condition $\mathbf{C}_{<T'}$ and target code sequences $\mathbf{C}_{\geq T', <j}$ with the code embedding matrix $\mathbf{W}^c$, and summing them along with the code ID dimension:

$$\mathbf{e}_t^c = \begin{cases} \sum_{k=0}^{7} \mathbf{W}^c \odot c_{t,k}, & t < T' \\ \sum_{k=0}^{j-1} \mathbf{W}^c \odot c_{t,k}, & t \geq T' \end{cases}, \tag{11}$$

where $t$ is the time step and $j$ is the codec code ID. Next, we obtain the codec code ID embedding $\mathbf{e}^j$ with the code ID embedding matrix $\mathbf{W}^{id}$.

$$\mathbf{e}^j = \mathbf{W}^{id} \odot j. \tag{12}$$

We concatenate the text embedding sequence $\mathbf{E}^x$, the code embedding sequence $\mathbf{E}^c$, and the codec code ID embedding $\mathbf{e}^j$, inserting the embedding of the special token $< eos >$ in the middle:

$$\mathbf{E}^j = \mathbf{E}^x \,\|\, [\mathbf{e}_{<\mathrm{eos}>}] \,\|\, \mathbf{E}^c \,\|\, [\mathbf{e}_{<\mathrm{eos}>}] \,\|\, [\mathbf{e}^j]. \tag{13}$$

We then separately add the learnable position embedding to the text embedding sequence and the code embedding sequence, similar to the AR model. The NAR model is fed with $\mathbf{E}^j$ and trained to predict the corresponding code sequence $\mathbf{c}_{:,j}$ for each codec code id $j$ using a code prediction layer. With the full attention mask strategy, the prediction of each token $c_{t,j}$ can attend to the entire input sequence, as depicted in the upper right part of Figure 2a.

Overall, the NAR model is optimized by minimizing the negative log likelihood of each $j$-th target code sequence $\mathbf{c}_{\geq T', j}$ conditioned on the text sequence $\mathbf{x}$, all the code sequences of the acoustic condition $\mathbf{C}_{<T'}$ and the preceding $j$ target code sequences $\mathbf{c}_{\geq T', <j}$.

$$\mathcal{L}_{\mathrm{NAR}} = -\log p(\mathbf{C}_{\geq T', \geq 1}|\mathbf{x}, \mathbf{C}_{<T'}, \mathbf{c}_{\geq T', 0}; \theta_{\mathrm{NAR}}) \tag{14}$$

$$= -\sum_{j=1}^{7} \log p(\mathbf{c}_{\geq T', j}|\mathbf{x}, \mathbf{C}_{<T'}, \mathbf{C}_{\geq T', <j}; \theta_{\mathrm{NAR}}). \tag{15}$$

---

**Algorithm 1** Repetition Aware Sampling in VALL-E 2 AR Model Decoding

---

1: **given** text condition $\mathbf{x}$, pre-trained AR model $\theta_{\mathrm{AR}}$, group size $G$, decoding step $t$, concatenation of code prompt and preceding group sequence $\mathbf{c}_{<t \cdot G,0}$, predicted code index $i$, top-p value $v$ for nucleus sampling, repetition threshold ratio $t_r$, window size $K$
2: infer the pre-trained AR model $\theta_{\mathrm{AR}}$ and predict the probability distribution $p(c_{t'}|\mathbf{x}, \mathbf{c}_{<t \cdot G,0}; \theta_{\mathrm{AR}})$
3: generate $c_{t'}$ by nucleus sampling from the probability distribution $p(c_{t'}|\mathbf{x}, \mathbf{c}_{<t \cdot G,0}; \theta_{\mathrm{AR}})$ with top-p value $v$
4: calculate the repetition ratio $r$ of the token $c_{t'}$ in the preceding code sequence with window size $K$:
$r \leftarrow \frac{1}{K} \sum_{k=0}^{K} 1_{c_{t'}=c_{t'-k}}$
5: **if** $r > t_r$ **then**
6:     replace $c_{t'}$ by random sampling from the probability distribution $p(c_{t'}|\mathbf{x}, \mathbf{c}_{<t \cdot G,0}; \theta_{\mathrm{AR}})$
7: **return** target code $c_{t'}$

---

In practice, to optimize computational efficiency during training, we do not calculate the training loss by iterating over all values of $j$ and aggregating the corresponding losses, but randomly select a $j \in [1, \ldots, 7]$ and optimize the model using the training loss:

$$\mathcal{L}_{\mathrm{NAR\_j}} = -\log p(\mathbf{c}_{\geq T',j}|\mathbf{x}, \mathbf{C}_{<T'}, \mathbf{C}_{\geq T',<j}; \theta_{\mathrm{NAR}}). \tag{16}$$

### 3.4 VALL-E 2 INFERENCE

Following VALL-E, we perform the zero-shot TTS task via prompting during inference. As depicted in Figure 2b, given the text sentence and the enrolled speech sample of the unseen speaker along with its corresponding transcription, we first concatenate the speech transcription and the text sentence, encoded into the text sequence $\mathbf{x}$ using the text tokenizer to serve as the text condition. The speech sample is converted into the codes $\mathbf{C}^P = \mathbf{C}_{<T'} = [\mathbf{c}_0, \mathbf{c}_1, \ldots, \mathbf{c}_{(T'-1)}]$ using the audio codec encoder to serve as the prompt. By prompting the conditional codec language model, we infer the AR model and NAR model to generate the target codes $\mathbf{C}_{\geq T'} = [\mathbf{c}_{T'}, \ldots, \mathbf{c}_{(T-1)}]$. Finally, the target codes is used by the audio codec decoder to synthesize the target personalized speech signals.

#### 3.4.1 AUTOREGRESSIVE MODEL INFERENCE

We first infer the AR model to generate the first code sequence of the target codes $\mathbf{c}_{\geq T',0}$ conditioned on the text sequence $\mathbf{x}$ and the code prompt $\mathbf{c}_{<T',0}$. With the grouped codec language modeling method, we feed the grouped code sequence to the AR model and generate each group of target codes in an autoregressive way:

$$\mathbf{c}_{\geq T',0} = \arg\max_{\mathbf{c}_{\geq T',0}} p(\mathbf{c}_{\geq T',0}|\mathbf{x}, \mathbf{c}_{<T',0}; \theta_{\mathrm{AR}}) \tag{17}$$

$$= \arg\max_{\mathbf{c}_{\geq T',0}} \sum_{t=T'/G}^{T/G-1} \sum_{t'=t \cdot G}^{(t+1) \cdot G-1} \log p(c_{t',0}|\mathbf{x}, \mathbf{c}_{<t \cdot G,0}; \theta_{\mathrm{AR}}). \tag{18}$$

Different from the random sampling method used in VALL-E, in this work, we propose a repetition aware sampling method to enhance nucleus sampling for the better decoding stability. As detailed in Algorithm 1, given the probability distribution $p(c_{t'}|\mathbf{x}, \mathbf{c}_{<t \cdot G,0}; \theta_{\mathrm{AR}})$ predicted by the AR model, we first generate the target code $c_{t'}$ by nucleus sampling with a pre-defined top-p value $v$. Then, we calculate the repetition ratio $r$ of token $c_{t'}$ in the preceding code sequence with a window size $K$. If the ratio $r$ exceeds a pre-defined repetition threshold ratio $t_n$, we replace the target code $c_{t'}$ by random sampling from $p(c_{t'}|\mathbf{x}, \mathbf{c}_{<t \cdot G,0}; \theta_{\mathrm{AR}})$. Although the codec codes in one group are modeled in a non-autoregressive way, they are predicted autoregressively so as to calculate the repetition ratio $r$ and switch between these two sampling methods. With this repetition aware sampling method, the decoding process can not only benefit from the stability of nucleus sampling, but also avoid the infinite loop issue with the help of random sampling. It should be noted that this repetition aware sampling won't increase the decoding latency since the runtime cost of the additional sampling operation is almost negligible compared to the model inference process.

#### 3.4.2 NON-AUTOREGRESSIVE MODEL INFERENCE

Given the first code sequence of the target codes $\mathbf{c}_{\geq T',0}$, we can infer the NAR model with the text condition $\mathbf{x}$ and the acoustic condition $\mathbf{C}_{<T'}$ to generate the remaining code sequences of the target

Table 1: Objective evaluation results on LibriSpeech test-clean.

| System | GroupSize | 3s Prefix as Prompt | | | Ref Utterance as Prompt | | |
|---|---|---|---|---|---|---|---|
| | | SIM↑ | WER↓ | DNSMOS↑ | SIM↑ | WER↓ | DNSMOS↑ |
| GroundTruth | - | 0.905 | 1.6 | 3.891 | 0.779 | 1.6 | 3.891 |
| ↪ Codec | - | 0.823 | 1.7 | 3.886 | 0.715 | 1.7 | 3.886 |
| *Single Sampling* | | | | | | | |
| VALL-E | 13ms | 0.773 | 2.3 | 3.942 | 0.633 | 3.1 | 3.985 |
| | ×1 | **0.782** | 1.6 | 3.947 | **0.643** | **1.5** | 3.987 |
| VALL-E 2 | ×2 | 0.777 | **1.5** | **3.966** | 0.635 | **1.5** | **4.000** |
| | ×4 | 0.773 | 1.8 | 3.950 | 0.615 | 2.2 | 3.967 |
| | ×8 | 0.766 | 2.5 | 3.937 | 0.566 | 4.2 | 3.875 |
| *Five-Time Sampling* | | | | | | | |
| VALL-E | 13ms | 0.802 | **1.0** | 3.944 | 0.676 | 0.8 | 3.987 |
| | ×1 | **0.807** | **1.0** | 3.943 | **0.687** | 0.7 | 3.994 |
| VALL-E 2 | ×2 | 0.803 | **1.0** | **3.967** | 0.679 | **0.6** | **3.997** |
| | ×4 | 0.799 | 1.1 | 3.954 | 0.662 | 0.7 | 3.973 |
| | ×8 | 0.790 | **1.0** | 3.938 | 0.616 | 1.0 | 3.898 |

codes $\mathbf{C}_{\geq T', \geq 1}$:

$$\mathbf{C}_{\geq T', \geq 1} = \arg\max_{\mathbf{C}_{\geq T', \geq 1}} p(\mathbf{C}_{\geq T', \geq 1} | \mathbf{x}, \mathbf{C}_{<T'}, \mathbf{c}_{\geq T', 0}; \theta_{\text{NAR}}) \qquad (19)$$

$$= \arg\max_{\mathbf{C}_{\geq T', \geq 1}} \sum_{j=1}^{7} \log p(\mathbf{c}_{\geq T', j} | \mathbf{x}, \mathbf{C}_{<T'}, \mathbf{C}_{\geq T', <j}; \theta_{\text{NAR}}). \qquad (20)$$

To generate the 2-8 code sequence, we perform inference on the NAR model seven times, generating them one by one using a greedy decoding method. Together with the first codec codes generated by the AR model, the whole code matrix $\mathbf{C}_{\geq T'}$ is used for generating the target personalized speech waveform with the corresponding audio codec decoder.

VALL-E 2 can not only use a reference utterance of an unseen speaker as prompt to generate the speech cloning his/her voice, but also be able to perform zero-shot speech continuation, in which, we use the complete transcription of the utterance as the text condition and the first 3-second prefix as the prompt for the target personalized speech generation.

## 4 EXPERIMENT

### 4.1 SETUPS

We use Libriheavy corpus (Kang et al., 2024) as the training data, and employ the same Transformer architecture as VALL-E for AR model and the NAR models in VALL-E 2. The open-sourced pre-trained Vocos model (Siuzdak, 2023) is served as the audio codec decoder for speech generation.

We use LibriSpeech test-clean (Panayotov et al., 2015) and VCTK (Veaux et al., 2016) for zero-shot TTS evaluation, ensuring none of the speakers from these corpora are included in the training data. We invite 20 external native American English speakers to evaluate the speaker similarity and comparative naturalness of the synthesized speech, quantified by Speaker Mean Opinion Score (SMOS) and Comparative Mean Opinion Score (CMOS), respectively. We also employ objective evaluation metrics including SIM, WER, and DNSMOS to assess speaker similarity, robustness, and overall perceived quality of each synthesized speech. For a better comparison in speech continuation, we evaluate the entire utterance instead of focusing solely on the continuation segment. In our experiment, we report the results of sampling once and five times for each speech synthesis. Please refer to Appendix A.1 and Appendix A.2 for further details on experimental setups and ablation studies of VALL-E 2.

### 4.2 LIBRISPEECH EVALUATION

#### 4.2.1 OBJECTIVE EVALUATION

Table 1 presents the objective evaluation results on the LibriSpeech test-clean dataset, where VALL-E 2 significantly outperforms VALL-E in all settings, even achieving better WER and DNSMOS scores than the ground truth speech with single sampling.

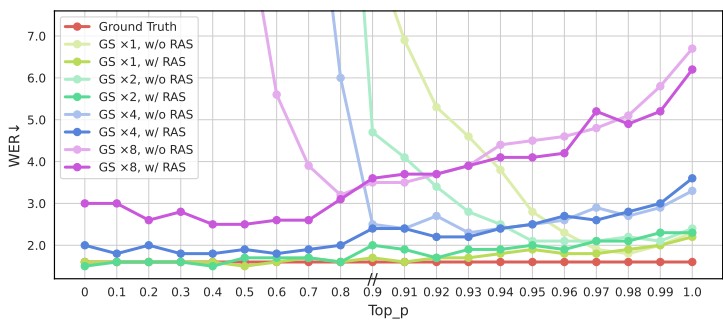

Figure 3: Decoding stability on LibriSpeech test-clean using 3s prefix as prompt. GS means group size and RAS stands for repetition aware sampling.

Table 2: Subjective evaluation results for 40 speakers on LibriSpeech test-clean, using a reference utterance as a prompt for each speaker.

| System | GroupSize | SMOS↑ | CMOS↑ |
|---|---|---|---|
| GroundTruth | - | $4.13_{\pm 0.32}$ | 0.00 |
| VALL-E | 13ms | $4.45_{\pm 0.28}$ | -0.268 |
| VALL-E 2 | ×1 | $\mathbf{4.61}_{\pm 0.19}$ | **0.033** |
| | ×2 | $4.51_{\pm 0.26}$ | -0.167 |

The SIM, WER, and DNSMOS scores of the ground truth speech are calculated as the upper bound. We observe a performance degradation in SIM and similar performance in WER and DNSMOS when using the off-the-shelf neural audio codec model for speech reconstruction. The baseline VALL-E can achieve impressive overall results with five-time sampling, but lack of robustness with single sampling, which could be attributed to the instability decoding process of random sampling.

In comparison, VALL-E 2 demonstrates significant improvement in robustness, especially in the single sampling scenario. With the repetition aware sampling, VALL-E 2 can successfully achieve better decoding stability, leads to the performance improvement in all the three metrics, and even obtain lower WER score than the ground truth speech. At times, a person may pronounce words unclearly, whereas a well-trained TTS system can avoid this issue. This demonstrates that our synthesized speech is highly faithful to both the provided text and the enrolled speaker's voice.

With the grouped code modeling, VALL-E 2 can achieve even better WER and DNSMOS scores with group size of 2 in the AR model. It demonstrates that this method can not only improve the inference efficiency by reducing the code sequence length, but also improve the model performance by mitigating the long context modeling problem. Even with group size of 4, we can still obtain similar or better results as the baseline model while greatly improve the inference efficiency by reducing the code sequence length by 4 times. Figure 3 further demonstrates the superior decoding stability of VALL-E 2. The repetition aware sampling method significantly enhances the decoding stability, regardless of the different group size setting. It enables VALL-E 2 to perform inference with a very small top-p (even 0), which tends to introduce much less errors and generate more robust speech codec codes than decoding with a large top-p. This is the key to obtaining a good WER score, even lower than that of ground truth speech, using a small top-p.

### 4.2.2 SUBJECTIVE EVALUATION

Table 2 presents the subjective evaluation results on the LibriSpeech test-clean. For the subjective evaluation, the previous utterance from the official speech list is used as the prompt to generate the current utterance for each speaker in the LibriSpeech test-clean dataset, resulting in 40 test cases.

As indicated in the table, VALL-E 2 can successfully surpasses VALL-E in terms of both speaker similarity SMOS and speech quality CMOS, even better performance than the ground truth speech. This suggests that our proposed method can achieve human parity zero-shot TTS performance in LibriSpeech benchmark. With group code modeling method, VALL-E 2 can also achieve better performance than VALL-E with group size of 2 for the inference of AR model.

Table 3: Objective evaluation results on VCTK.

| System | GroupSize | 3s Prompt | | | 5s Prompt | | | 10s Prompt | | |
|---|---|---|---|---|---|---|---|---|---|---|
| | | SIM↑ | WER↓ | DNSMOS↑ | SIM↑ | WER↓ | DNSMOS↑ | SIM↑ | WER↓ | DNSMOS↑ |
| GroundTruth | - | 0.623 | 0.3 | 3.635 | 0.679 | 0.3 | 3.635 | 0.709 | 0.3 | 3.635 |
| ↪ Codec | - | 0.563 | 0.3 | 3.609 | 0.616 | 0.3 | 3.609 | 0.644 | 0.3 | 3.609 |
| *Single Sampling* | | | | | | | | | | |
| VALL-E | 13ms | 0.430 | 2.4 | **3.667** | 0.455 | 3.1 | 3.664 | 0.533 | 5.8 | 3.575 |
| VALL-E 2 | ×1 | **0.447** | **0.9** | 3.666 | **0.487** | 1.9 | **3.674** | **0.558** | 3.3 | **3.667** |
| | ×2 | 0.426 | 1.5 | 3.599 | 0.481 | **0.9** | 3.598 | 0.557 | 2.3 | 3.617 |
| | ×4 | 0.417 | 1.8 | 3.470 | 0.457 | 2.1 | 3.537 | 0.521 | 2.9 | 3.547 |
| | ×8 | 0.375 | 5.0 | 3.438 | 0.415 | 4.8 | 3.387 | 0.499 | 8.0 | 3.420 |
| *Five-Time Sampling* | | | | | | | | | | |
| VALL-E | 13ms | 0.497 | 0.3 | 3.599 | 0.534 | 0.3 | 3.666 | 0.607 | 1.5 | 3.591 |
| VALL-E 2 | ×1 | **0.508** | **0.0** | **3.684** | **0.552** | 0.3 | **3.699** | **0.620** | 1.5 | **3.694** |
| | ×2 | 0.494 | 1.0 | 3.616 | 0.547 | **0.1** | 3.617 | 0.606 | **0.4** | 3.621 |
| | ×4 | 0.487 | 0.9 | 3.547 | 0.531 | 0.4 | 3.588 | 0.592 | 1.6 | 3.559 |
| | ×8 | 0.444 | 2.4 | 3.454 | 0.499 | 0.5 | 3.429 | 0.563 | 1.3 | 3.430 |

Table 4: Subjective evaluation results for 60 speakers on VCTK.

| System | GroupSize | 3s Prompt | | 5s Prompt | | 10s Prompt | |
|---|---|---|---|---|---|---|---|
| | | SMOS↑ | CMOS↑ | SMOS↑ | CMOS↑ | SMOS↑ | CMOS↑ |
| GroundTruth | - | $4.47_{\pm 0.13}$ | 0.00 | $4.53_{\pm 0.14}$ | 0.00 | $4.74_{\pm 0.17}$ | 0.00 |
| VALL-E | 13ms | $4.32_{\pm 0.16}$ | **0.028** | $4.05_{\pm 0.20}$ | **0.144** | $3.50_{\pm 0.49}$ | **0.094** |
| VALL-E 2 | ×1 | $4.42_{\pm 0.15}$ | **0.207** | $4.28_{\pm 0.16}$ | 0.079 | $3.95_{\pm 0.10}$ | **0.117** |
| | ×2 | **$4.47_{\pm 0.13}$** | **0.163** | $4.14_{\pm 0.17}$ | **0.217** | $4.26_{\pm 0.42}$ | **0.109** |

## 4.3 VCTK EVALUATION

### 4.3.1 OBJECTIVE EVALUATION

Table 3 presents the objective evaluation results on the VCTK dataset, where VALL-E 2 demonstrates superior zero-shot TTS performance than VALL-E, especially in terms of speech robustness score WER. It demonstrates the repetition aware sampling method can also effectively stable the decoding process on challenging VCTK data with speakers in diverse accents. It can roughly half the WER score in the single sampling scenario. With five-time sampling, we can effectively filter out low-quality samples and select the best sample as the output, enabling VALL-E to generate speech of much better robustness, and mitigate the gap of the WER score between VALL-E and VALL-E 2.

When comparing different prompt lengths, we find that the grouped code modeling method can even further improve the WER score for longer prompts. The reason could be that the excessively long prompts present challenges in the long sequence modeling of the Transformer architecture and tend to yield some generation errors due to incorrect attention alignments, and the grouped code modeling method can alleviate this problem by reducing the sequence length while enhancing the AR modeling.

### 4.3.2 SUBJECTIVE EVALUATION

Table 4 presents the subjective evaluation results on the VCTK dataset. We conduct the subjective evaluation with 60 test cases from 60 distinct speakers. Given the diverse speaker accents in the VCTK dataset, zero-shot TTS is much more challenging than that on LibriSpeech dataset. The comparison result in Table 4 reveals that VALL-E 2 can successfully surpasses VALL-E in terms of both speaker similarity and speech quality, even same or better performance than the ground truth speech when using only 3s prompt. This underscores the human parity performance of VALL-E in zero-shot TTS for a very diverse accents scenario. Thanks to the long context modeling capability of group code modeling method, we also achieve significant performance improvement with long prompt of 10s, especially for speaker similarity.

## 5 CONCLUSION

We introduce VALL-E 2, a language modeling approach that achieves human parity zero-shot text to speech synthesis for the first time. Based on the success of VALL-E, VALL-E 2 introduce two simple but effective methods: repetition aware sampling for better decoding stability and grouped code modeling for better modeling efficiency. Furthermore, our observations reveal that VALL-E 2 is capable of reliably synthesizing speech for complex sentences, including those that are challenging to read or contain numerous repeated phrase.

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

# A APPENDIX

## A.1 EXPERIMENTAL SETUPS

### A.1.1 MODEL TRAINING

We use Libriheavy corpus (Kang et al., 2024) as the training data. This corpus is a labeled version of the Librilight corpus (Kahn et al., 2020) that contains 50k hours of speech with around 7000 distinct speakers derived from open-source English audiobooks that are part of the LibriVox project[1]. We use Byte-Pair Encoding (BPE) for text tokenization, and the pre-trained open-sourced EnCodec model (Défossez et al., 2022) at 6K bitrates for 24kHz audio reconstruction for speech tokenization. Additionally, we use the open-sourced pre-trained Vocos model (Siuzdak, 2023) as the audio codec decoder for speech generation.

Following VALL-E, both the AR model and the NAR models employ the same Transformer architecture in VALL-E 2. In our experiments, we mainly evaluate 4 VALL-E 2 models, which share the same NAR model but different AR models. The 4 AR models corresponds to the group size of 1, 2, 4 and 8. Among these models, the AR model with group size of 1 is implemented without the group embedding layer and group prediction layer, and the baseline model VALL-E employs the same NAR model and AR model with group size of 1[2].

Both the AR and NAR models are trained using 16 NVIDIA TESLA V100 32GB GPUs. The models are optimized with the AdamW optimizer, with the learning rate warmed up for the first 32k updates to a peak of learning rate, then linearly decayed. For NAR model training, the length of the acoustic condition is randomly sampled to be the maximum of half of the current utterance with a random value from 3s to 30s.

### A.1.2 EVALUATION METRICS

We employ subjective evaluation metrics, including SMOS and CMOS, to assess the speaker similarity and comparative naturalness of synthesized speech, respectively. We invite 20 external native speakers of American English to participate as contributors in a crowdsourcing effort to evaluate each speech from various perspectives.

**SMOS** (Similarity Mean Opinion Score) is used to evaluate the speaker similarity of the speech to the original prompt. The SMOS scale ranges from 1 to 5, with increments of 0.5 points.

**CMOS** (Comparative Mean Opinion Score) is used to evaluate the comparative naturalness of the synthesized speech against a given reference speech. The CMOS scale ranges from -3 (indicating the synthesized speech of the new system is much worse than the reference) to 3 (indicating the new system is much better than the reference), with intervals of 1. In our study, we use the ground truth speech as the comparison reference.

---

[1]https://librivox.org

[2]We re-train the baseline VALL-E model with the Libriheavy dataset for fair comparison.

We also employ objective evaluation metrics including SIM, WER, and DNSMOS to assess the speaker similarity, robustness, and overall perceived quality of each synthesized speech. For a better comparison in speech continuation, we evaluate the entire utterance instead of focusing solely on the continuation segment.

**SIM** is used to evaluate the speaker similarity between the original prompt and synthesized speech, leveraging the SOTA speaker verification model, WavLM-TDNN [3] (Chen et al., 2022). The similarity score predicted by WavLM-TDNN is in the range of $[-1, 1]$, with a larger value indicating higher speaker similarity.

**WER** (Word Error Rate) is used to evaluate the robustness of synthesized speech. Neural TTS systems sometimes experience deletion, insertion, and replacement errors due to incorrect attention alignments, which can affect their robustness. We perform ASR on the generated audio and calculate the WER with respect to the original transcriptions. In this experiment, we employ the open-sourced Conformer-Transducer model[4] (Gulati et al., 2020) as the ASR model.

**DNSMOS** (Deep Noise Suppression Mean Opinion Score) is used to assess the overall perceived quality of the generated speech (Reddy et al., 2021). Specifically, we use a model trained with ground truth human ratings obtained using ITU-T P.808 (ITU, 2018)[5] to predict the DNSMOS score, which is in the range of $[1, 5]$, with a larger value indicating better quality.

### A.1.3 EVALUATION SETTINGS

We use LibriSpeech test-clean (Panayotov et al., 2015) and VCTK (Veaux et al., 2016) for zero-shot TTS evaluation, ensuring none of the speakers from these corpora are included in the training data.

**LibriSpeech** test-clean is an official test split from the LibriSpeech corpus, containing English speech sampled at 16kHz. It originates from the same domain of the LibriVox project as the training data but features different speaker IDs. Following Borsos et al. (2022) and Wang et al. (2023a), we use samples from LibriSpeech test-clean with lengths between 4 and 10 seconds, resulting in a 2.2 hours subset and 40 unique speakers. We evaluate each sample synthesis under two settings: 3s Prefix as Prompt and Ref Utterance as Prompt. For the first setting, we perform speech continuation and utilize the 3-second prefix of the speech as the prompt. In the second setting, we use a reference utterance from the same speaker as the prompt. Specifically, we begin by filtering the official speech list of LibriSpeech test-clean based on length. For the ordered speech list of each speaker, in the first setting, we synthesize the $i$-th speech sample using the first 3 seconds of the ground-truth $i$-th speech sample as the prompt. In the second setting, we synthesize the $i$-th speech sample using the $(i-1)$-th sample as the prompt and synthesize the first speech sample using the last sample as the prompt.

**VCTK** is a reading corpus with speech sampled at 48kHz by 108 English speakers. Compared to LibriSpeech, VCTK presents a greater challenge as it encompasses speakers with a wide range of accents. We evaluate each sample synthesis under three settings: using prompts of 3s, 5s, and 10s in length. Specifically, for each speaker, we select an utterance whose length is closest to but less than 3s/5s/10s to serve as the prompts. We then randomly sample another utterance and use the corresponding transcription as the text input for speech synthesis.

For each sample synthesis, we first perform inference with the AR model to generate the first code sequence using the repetition aware sampling method (Section 3.4.1), where we set the hyperparameter $K = 10$, $t_r = 0.1$, and select the top-p value $v$ from 0.0 to 0.8 with the intervals of 0.1. Next, we perform inference on the NAR model seven times to generate the remaining seven code sequences using a greedy decoding method. The sampling-based decoding method of the AR model allows us to generate diverse samples from the same input.

In our experiment, we report the results of sampling once and five times for each speech synthesis. For the five-time sampling, we report the results of sorting on SIM and WER: We sort the samples based on the speaker similarity and robustness scores, represented by the SIM and WER scores. Specifically, given the five samples $\{\hat{\mathbf{y}}_i\}_{i=1}^5$ with the corresponding SIM, WER, and DNSMOS scores denoted as $\hat{\mathbf{y}}_i^{\text{SIM}}$, $\hat{\mathbf{y}}_i^{\text{WER}}$, and $\hat{\mathbf{y}}_i^{\text{DNSMOS}}$, we sort them according to the WER score if the SIM

---

[3]We use the best speaker verification model released at `https://github.com/microsoft/UniSpeech/tree/main/downstreams/speaker_verification#pre-trained-models`

[4]`https://huggingface.co/nvidia/stt_en_conformer_transducer_xlarge`

[5]`https://github.com/microsoft/DNS-Challenge/tree/master/DNSMOS`

Table 5: Ablation study of model input on LibriSpeech test-clean. The symbol ◆ denotes that the acoustic condition is not explicitly split during the NAR model training, and the prompt is treated as the prefix of the target code matrix during the NAR model inference.

| AR Model | NAR Model | | 3s Prefix as Prompt | | | Ref Utterance as Prompt | | |
|---|---|---|---|---|---|---|---|---|
| Prompt Input | Text Input | Prompt Input | SIM↑ | WER↓ | DNSMOS↑ | SIM↑ | WER↓ | DNSMOS↑ |
| *Single Sampling* | | | | | | | | |
| ✓ | ✓ | ✓ | 0.779 | 1.6 | 3.956 | 0.639 | 1.9 | 4.013 |
| ✗ | ✓ | ✓ | n/a | n/a | n/a | 0.169 | 2.8 | 4.001 |
| ✓ | ✓ | ◆ | 0.731 | 1.6 | 3.957 | 0.530 | 1.9 | 4.018 |
| ✓ | ✓ | ✗ | n/a | n/a | n/a | 0.385 | 1.8 | 4.015 |
| ✓ | ✗ | ✓ | 0.774 | 5.6 | 3.958 | 0.619 | 10.0 | 4.016 |
| *Five-Time Sampling* | | | | | | | | |
| ✓ | ✓ | ✓ | 0.804 | 1.0 | 3.952 | 0.684 | 0.7 | 4.016 |
| ✗ | ✓ | ✓ | n/a | n/a | n/a | 0.305 | 2.0 | 4.018 |
| ✓ | ✓ | ◆ | 0.765 | 1.0 | 3.956 | 0.583 | 0.7 | 4.020 |
| ✓ | ✓ | ✗ | n/a | n/a | n/a | 0.457 | 1.0 | 4.019 |
| ✓ | ✗ | ✓ | 0.793 | 1.8 | 3.960 | 0.647 | 3.0 | 4.018 |

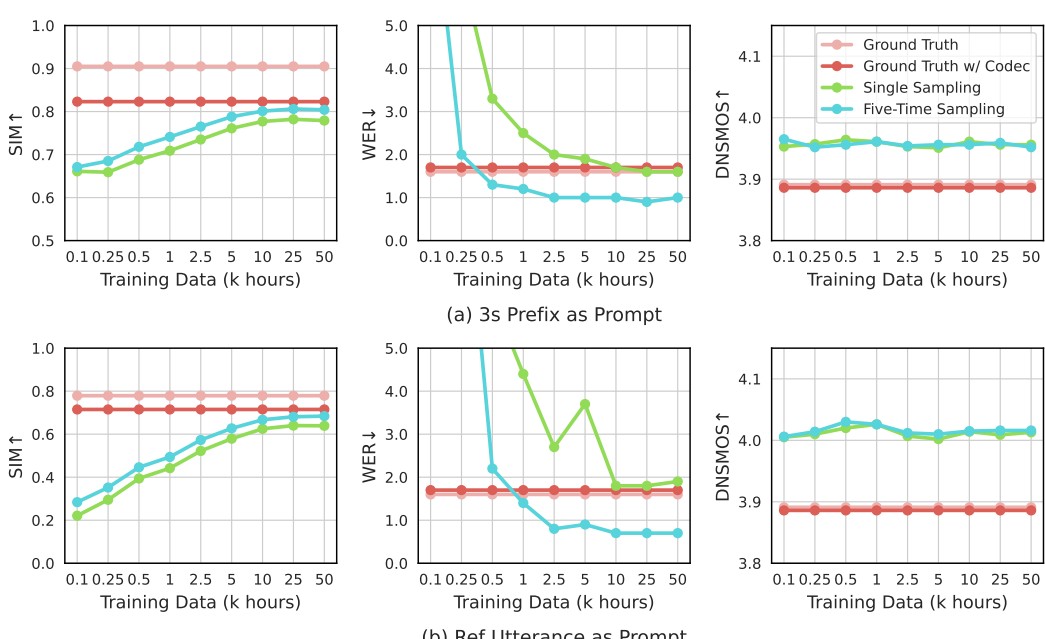

Figure 4: Ablation study of the size of training data on LibriSpeech test-clean.

score is greater than 0.3 and sort according to the SIM score otherwise. This sorting method can be expressed as:

$$\hat{\mathbf{y}}_{\text{best}} = \arg\max_{\hat{\mathbf{y}}_i}([\min(\hat{\mathbf{y}}_i^{\text{SIM}}, 0.3), 1 - \hat{\mathbf{y}}_i^{\text{WER}}]), \qquad (21)$$

where $\max(\cdot)$ denotes finding the lexicographically largest array [6]. The resulting SIM, WER, and DNSMOS scores are $\hat{\mathbf{y}}_{\text{best}}^{\text{SIM}}$, $\hat{\mathbf{y}}_{\text{best}}^{\text{WER}}$ and $\hat{\mathbf{y}}_{\text{best}}^{\text{DNSMOS}}$.

Table 6: Ablation study of model input on VCTK.

| AR Model | NAR Model | | 3s Prompt | | | 5s Prompt | | | 10s Prompt | | |
|---|---|---|---|---|---|---|---|---|---|---|---|
| Prompt Input | Text Input | Prompt Input | SIM↑ | WER↓ | DNSMOS↑ | SIM↑ | WER↓ | DNSMOS↑ | SIM↑ | WER↓ | DNSMOS↑ |
| | | | | | *Single Sampling* | | | | | | |
| ✓ | ✓ | ✓ | 0.450 | 2.6 | 3.698 | 0.486 | 2.0 | 3.692 | 0.567 | 4.1 | 3.684 |
| ✗ | ✓ | ✓ | 0.139 | 3.0 | 3.685 | 0.144 | 2.9 | 3.686 | 0.159 | 3.5 | 3.672 |
| ✓ | ✓ | ◆ | 0.347 | 2.3 | 3.684 | 0.396 | 2.4 | 3.672 | 0.489 | 4.4 | 3.688 |
| ✓ | ✓ | ✗ | 0.224 | 2.3 | 3.686 | 0.245 | 2.4 | 3.679 | 0.284 | 3.8 | 3.690 |
| ✓ | ✗ | ✓ | 0.426 | 14.1 | 3.698 | 0.478 | 11.9 | 3.705 | 0.556 | 11.5 | 3.677 |
| | | | | | *Five-Time Sampling* | | | | | | |
| ✓ | ✓ | ✓ | 0.513 | 0.0 | 3.678 | 0.550 | 0.0 | 3.694 | 0.618 | 1.6 | 3.703 |
| ✗ | ✓ | ✓ | 0.271 | 1.6 | 3.787 | 0.282 | 2.3 | 3.741 | 0.303 | 3.1 | 3.725 |
| ✓ | ✓ | ◆ | 0.418 | 0.4 | 3.665 | 0.472 | 1.0 | 3.700 | 0.550 | 1.5 | 3.675 |
| ✓ | ✓ | ✗ | 0.306 | 1.4 | 3.658 | 0.327 | 2.1 | 3.678 | 0.361 | 3.8 | 3.677 |
| ✓ | ✗ | ✓ | 0.476 | 3.0 | 3.705 | 0.527 | 1.5 | 3.719 | 0.605 | 2.4 | 3.725 |

## A.2 ABLATION STUDY

### A.2.1 LIBRISPEECH

We conduct several ablation studies of VALL-E 2 on LibriSpeech test-clean. We use the VALL-E 2 model with group size 1, and present the results for both single-sampling and five-time sampling for each speech synthesis. For five-time sampling, we select the best candidate by sorting 5 samples based on SIM and WER scores as in Equation 21.

**Ablation on Model Input**: In Table 5, we study the impact of the text and prompt input in the AR and NAR models. Removing the prompt in either AR or NAR model results in significantly lower speaker similarity scores, emphasizing the crucial role of the prompt in preserving speaker identity. Despite the NAR model having access to the prompt, the AR model's prompt still contributes significantly to speaker similarity. In the case of the NAR model, we also discover that explicitly splitting the acoustic condition during training is essential to enhance the final speaker similarity score, as the NAR model can extract more speaker information from the entire 8 code sequences of the prompt. Interestingly, we find that the prompt in the AR model also improves the robustness of the generated speech, as evidenced by a lower WER score. This can be attributed to the prompt's ability to constrain the search space of the one-to-many speech synthesis task, thereby enabling more stable and robust speech generation. Additionally, the text input is also crucial in the NAR model for achieving a lower WER score, despite its use in the AR model.

**Ablation on Training Data**: In Figure 4, we explore the impact of the size of training data on the zero-shot TTS performance. We find that our model, with 10k training data, can already achieve performance similar to that with 50k training data on LibriSpeech test-clean. The additional 40k data only results in slight performance improvement in terms of speaker similarity and robustness. However, if we reduce the training data to less than 10k, we observe a performance degradation, especially for the setting of reference utterance as a prompt. It should be noted that this conclusion is based on the current experiment setting in the audiobook domain.

### A.2.2 VCTK

We further conduct ablation studies of VALL-E 2 on VCTK dataset. We use the VALL-E 2 model with group size 1, and present the results for both single-sampling and five-time sampling for each speech synthesis. For five-time sampling, we sort multiple samples with Equation 21.

**Ablation on Model Input**: As shown in Table 6, consistent with the observations in the LibriSpeech evaluation, the prompt is crucial in both AR and NAR models for speaker information modeling. The speaker similarity score would significantly declines when we remove the prompt input. Although the text input is consumed in the AR model, the NAR model also requires it to synthesize robust speech.

**Ablation on Training Data**: As shown in Figure 5, the optimal size of training data varies for different inference prompts and metrics. The SIM score consistently benefits from larger training data, which offers more diverse speaker voice patterns. The best WER score with a 3s prompt requires

---

[6]Lexicographic order: given two partially ordered sets $A$ and $B$, the lexicographical order on the Cartesian product $A \times B$ is defined as $[a, b] \leq [a', b']$ if and only if $a < a'$ or $(a = a'$ and $b \leq b')$.

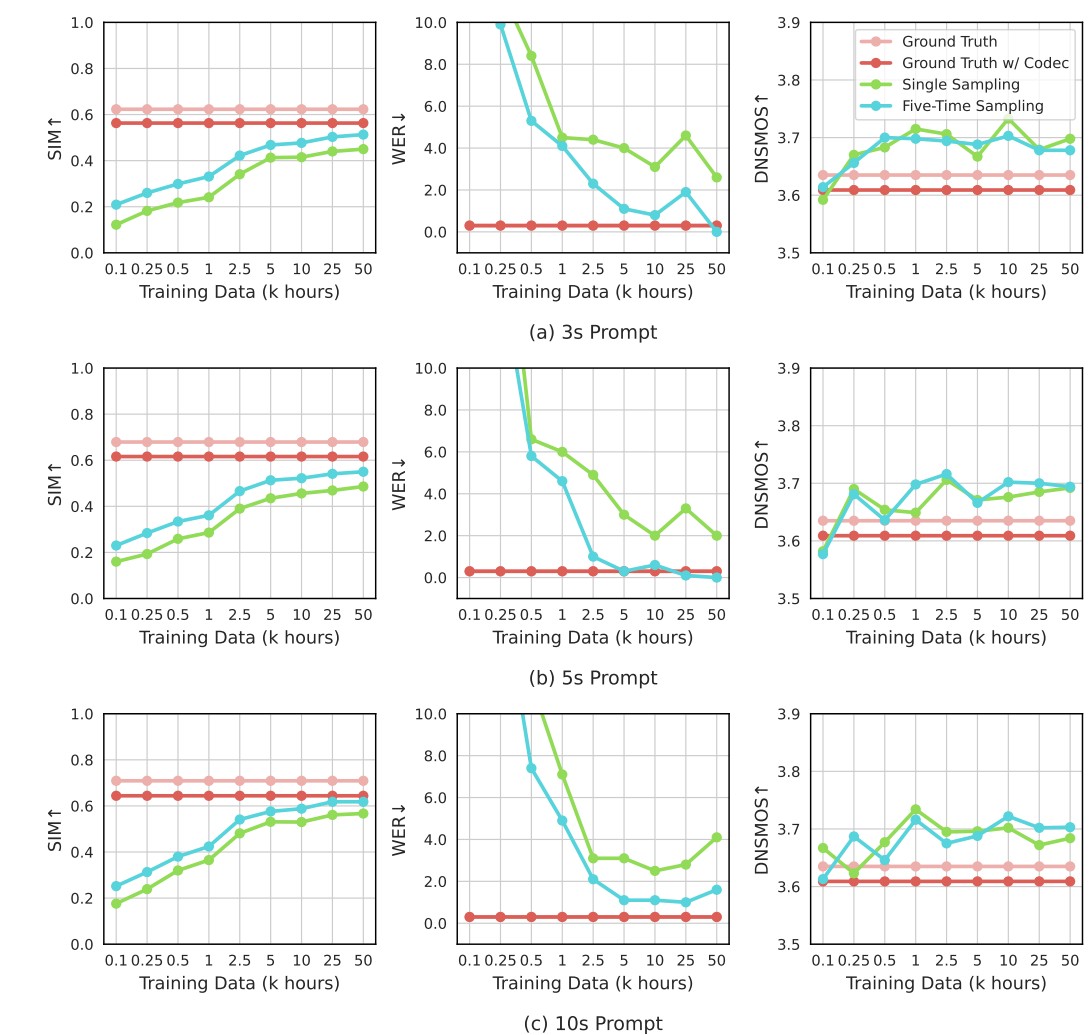

Figure 5: Ablation study of the size of training data on VCTK.

more training data than the 5s prompt and 10s prompt, due to the increased challenge of zero-shot TTS with only a 3s enrolled speech. Interestingly, the best DNSMOS score is not achieved with the largest training data. A possible explanation is that, with limited model capacity, our model achieves better speaker similarity and robustness at the expense of slight losses in perceived quality.

