# OpenReview forum: "VALL-E 2: Neural Codec Language Models are Human Parity Zero-Shot Text to Speech Synthesizers"
_ICLR.cc/2025/Conference — Submitted to ICLR 2025_

### Official Review · Reviewer_rff7 · 2024-10-18

**Soundness:** 3
**Presentation:** 3
**Contribution:** 3
**Rating:** 6
**Confidence:** 4

**Summary:**

This work is an extension of VALL-E, which aims to solve its two problems. 1. inference repetitions --> degrade performance 2. too long codec sequence for modeling --> degrade speed.

Specifically, they propose 1. Repetition Aware Sampling to remove repititions by accounting for token repetition in the decoding history, thus improve the synthesis quality, 2. Grouped Code Modeling to re-organize codec sequence into groups to shorten the length, thus improve the modeling efficiency.

From my side, it is a good extension of VALL-E series and solve its practical issues. But from research perpective, this work does not convey much novelty or insights, especially given the high requirement of ICLR conference.

**Strengths:**

see above

**Weaknesses:**

see above

**Questions:**

NA

---

### Official Review · Reviewer_gHCZ · 2024-10-23

**Soundness:** 4
**Presentation:** 4
**Contribution:** 4
**Rating:** 8
**Confidence:** 5

**Summary:**

VALL-E represents a breakthrough in neural codec language modeling for zero-shot text-to-speech synthesis. It can synthesize personalized speech from just a 3-second recording, while preserving the speaker's voice, emotion, and acoustic environment. VALL-E uses an autoregressive transformer to model coarse codec codes (1st group of EnCodec) and a non-autoregressive transformer to generate fine codec codes (2nd-8th groups of EnCodec). However, VALL-E faces two key limitations: 1) Stability: Random sampling during inference can cause instability, while small top-p nucleus sampling risks infinite loops. 2) Efficiency: Its autoregressive architecture is constrained by a fixed high frame rate, slowing inference.

The paper introduces VALL-E 2, which addresses the aforementioned issues with two innovations: Repetition Aware Sampling, which stabilizes decoding without increasing computational costs, and Grouped Code Modeling, which reduces sequence length and speeds up inference. These improvements make VALL-E 2 more robust, natural, and efficient in zero-shot TTS, achieving human parity for the first time on benchmarks including LibriSpeech and VCTK. VALL-E 2 can stably generate high-quality speech for complex sentences that are hard to read or contain many repeated phrases.

Considering that VALL-E (pre-printed on January 5, 2023) has not been published in any conference or journal, yet has already garnered 542 citations and opened the field for neural codec models, and that VALL-E 2 builds on this foundation by achieving human parity for the first time, I believe the combining contributions makes it deserving of acceptance.

**Strengths:**

- VALL-E 2 achieving human parity in zero-shot TTS is a promising advancement, marking a new benchmark for text-to-speech systems. Its potential applications are particularly promising in assistive technologies for individuals with speech impairments.

- The introduction of repetition-aware sampling and grouped code modeling is a simple but effective approach that enhances the model's stability and efficiency in generating speech. These methods could be easily adapted to speech-to-speech language models.

- The paper demonstrates strong experimental validation with comprehensive evaluations on datasets including LibriSpeech and VCTK, showing clear improvements in robustness, naturalness, and speaker similarity​.

- The technical explanations and results are clearly presented, making the contributions and performance enhancements easy to understand.

**Weaknesses:**

See above

**Questions:**

- Why do this paper choose Byte-Pair Encoding (BPE) for text tokenization instead of using phonemes? How many BPE tokens are used in the model? Given that large datasets like LibriHeavy typically require thousands of BPE classes, while phoneme-based tokenization usually involves only a few dozen classes, how do you anticipate this choice impacts the model’s performance?

- Including punctuation marks in modeling units could benefit text-to-speech (TTS) systems. I’m interested to know if the BPE units in this work incorporate punctuation marks. How might this decision impact the model’s performance?

---

### Official Review · Reviewer_cpdk · 2024-10-28

**Soundness:** 3
**Presentation:** 3
**Contribution:** 2
**Rating:** 3
**Confidence:** 4

**Summary:**

The authors proposed a neural codec language model for zero-shot text-to-speech, enhancing robustness by refining sampled tokens through repetition-aware sampling. They further improved robustness and efficiency by applying grouped code modeling, effectively reducing sequence length.

**Strengths:**

They enhanced the baseline model, VALL-E, the first neural codec language model, by introducing repetition-aware sampling and grouped code modeling. While the baseline models are prone to issues like word repetition or omission, the proposed methods mitigate these problems and further improve model efficiency.

**Weaknesses:**

1.	[Grouped Code Modeling1] From an engineering perspective for neural codec language models, the proposed grouped code modeling could improve model performance and efficiency. However, grouped code modeling is already a well-known technique in language models, as seen in works like [MegaByte], [RQ-Transformer], and [Block Transformer].

[MegaByte] Yu, Lili, et al. "Megabyte: Predicting million-byte sequences with multiscale transformers." Advances in Neural Information Processing Systems 36 (2023): 78808-78823.

[RQ-Transformer] Lee, Doyup, et al. "Autoregressive image generation using residual quantization." Proceedings of the IEEE/CVF Conference on Computer Vision and Pattern Recognition. 2022.

[Block Transformer] Ho, Namgyu, et al. "Block Transformer: Global-to-Local Language Modeling for Fast Inference." arXiv preprint arXiv:2406.02657 (2024).

2.	[Grouped Code Modeling2] Additionally, [UniAudio] and [GPST] have already adopted a similar structure to sample RVQ tokens more efficiently. While there may be slight differences in implementation, they have the same goal

[UniAudio] Yang, Dongchao, et al. "UniAudio: Towards Universal Audio Generation with Large Language Models." Forty-first International Conference on Machine Learning.

[GPST] Zhu, Yongxin, et al. "Generative Pre-trained Speech Language Model with Efficient Hierarchical Transformer." ACL, 2024.

3.	[Grouped Code Modeling3] Notably, the classic sequence-to-sequence text-to-speech Tacotron also utilized the grouped spectrogram sampling through a reduction factor.

[Tacotron] Wang, Yuxuan, et al. "Tacotron: Towards end-to-end speech synthesis." arXiv preprint arXiv:1703.10135 (2017).

4.	[Grouped Code Modeling4] The recently proposed model [MELLE] also claims to predict multiple frames per step, accelerating inference and mitigating robustness issues associated with long-sequence modeling while maintaining strong performance. Moreover, MELLE has been shown to outperform VALL-E2. This hurts the contribution of the proposed method.

5.	[Repetition Aware Sampling] Recently, Flow-matching and MaskGIT-based text-to-speech models have adopted iterative sampling methods similar to repetition-aware sampling. It would be beneficial for the authors to discuss and compare repetition-aware sampling with these iterative methods, particularly against models like VoiceBox and E2-TTS.Specifically, I hope to see the comparison with VoiceBox and E2-TTS.

6.	[Weak Baseline] The authors only compared the model with VALL-E. However, VALL-E underperforms compared to VoiceBox, E2-TTS, DiTTo-TTS, and CosyVoice.

I sincerely acknowledge the novel contribution of VALL-E in opening the door for neural codec language models; however, the novelty of VALL-E2 does not meet the standards expected for ICLR.

**Questions:**

[Q1. Comparison with Low-bitrate Codec] Have you compared the grouped Code Modeling with low-bitrate Codec?

---

### Official Review · Reviewer_DjJg · 2024-10-29

**Soundness:** 2
**Presentation:** 2
**Contribution:** 2
**Rating:** 3
**Confidence:** 5

**Summary:**

VALL-E 2 is an LM-based TTS model based on VALL-E. It proposes two new methods:

1. **Repetition Aware Sampling**: In this method, during the sampling process, the repetition ratio is calculated based on the number of times a token has been generated. If this value exceeds a threshold, tokens are generated randomly from the original distribution.\\
2. **Grouped Code Modeling**: This method reduces sequence length by grouping adjacent tokens into fixed-size groups.

Thanks to these contributions, VALL-E 2 achieves significantly higher performance than the baseline VALL-E, particularly yielding better subjective evaluation results than the ground truth on LibriSpeech test-clean and VCTK.

**Strengths:**

The paper is written in a clear and accessible manner, enhancing readability and comprehension. Notably, VALL-E 2 demonstrates superior performance over ground truth in subjective evaluations, achieving higher scores in both CMOS and SMOS on both the LibriSpeech test-clean and VCTK datasets.

**Weaknesses:**

The authors have made an effort to present this work as a promising study; however, upon closer examination, there are numerous concerns that require substantial improvement.

- **On Subjective Evaluation Results**:
  The results discussed as a strength in Figure 1 are fundamentally flawed due to the dataset disparity with other studies (e.g., NaturalSpeech3 [1] uses the Librilight dataset). This undermines fairness in comparison. To ensure meaningful comparison, the authors should replicate NaturalSpeech3, which currently appears to be a good model, and train on the same dataset.

- **On Grouped Code Modeling**:
  While this approach has some merit as a method to reduce the sequence length given the codec model’s high 75Hz frequency, it is rather naïve and cannot be considered innovative. In fact, similar efforts have already been undertaken in existing research, such as [2], which the authors should have cited at minimum. Additionally, the method does not lead to significant improvements in either objective or subjective evaluations, suggesting that further refinements are needed.

- **On Repetition Aware Sampling**:
  Although this method appears to address the traditional issue of repetition in models like VALL-E effectively, it is not particularly innovative. In language modeling (LLM) contexts, penalties for repetition have long been in use [3], making the lack of reference to these approaches surprising. While the authors’ method differs slightly from these established approaches, it would be necessary to compare with them to clarify the method’s effectiveness. Moreover, the existing application of repetition penalties in TTS contexts, as seen in [4], further accentuates this concern.

- **On Ablation Studies**:
  There is a significant lack of ablation studies. The paper includes excessive unnecessary information; for example, the equations related to the model are redundant, and condensing this information would allow the inclusion of ablation studies directly in the main text. The limited experiments in the appendix also lack relevance. Ablations such as the presence or absence of prompts and dataset size variations are not particularly noteworthy, and their results are self-evident. More critical studies, such as comparisons with traditional repetition penalties or ablations involving Vocos (a major change from VALL-E), would have been more appropriate.

- **On Baseline Comparisons**:
  Changing the decoder from VALL-E’s original to Vocos represents a major shift and warrants stronger emphasis in comparative experiments. Additionally, the fact that subjective evaluation is best when the group size is 1 makes it very challenging to establish differentiation from the baseline.

- **Contribution to the Field**:
  The lack of code and weight release significantly diminishes the contribution of this study to the field.


[1]: Ju, Zeqian, et al. "Naturalspeech 3: Zero-shot speech synthesis with factorized codec and diffusion models." ICML 2024.\
[2]: Tang, Changli, et al. "Salmonn: Towards generic hearing abilities for large language models." ICLR 2024.\
[3]: Keskar, Nitish Shirish, et al. "Ctrl: A conditional transformer language model for controllable generation." arXiv preprint arXiv:1909.05858 (2019).\
[4]: Casanova, Edresson, et al. "XTTS: a Massively Multilingual Zero-Shot Text-to-Speech Model." INTERSPEECH 2024.

**Questions:**

- **Why was Vocos selected?** Given that other Vocoders with potentially better performance are available, why was Vocos specifically chosen? Additionally, what was the necessity of switching the decoder from Encodec’s original model?

- **Reason for the Significant Improvement in SIM**: It is understandable that Repetition Aware Sampling could lead to an improvement in WER; however, it is less clear how this would directly impact SIM. Furthermore, why does the subjective evaluation show significant improvement despite relatively poor objective metrics?

- **Lack of Evaluation on Difficult Cases**: The introduction references challenging cases, yet no evaluation related to these cases is provided. Why is this evaluation absent?

---

### Meta-Review · Area_Chair_SmMy · 2024-12-18

**Metareview:**

The paper proposes to incorporate a new sampling procedure and a different grouping of codes to improve VALL-E for zero-shot TTS.

I recommend a rejection because all reviewers unanimously agree that paper lacks novelty despite the strong performance.

There are various other suggestions provided by the reviewers that the authors should consider to improve the paper. In particular, the comparison across multiple systems should be done carefully to avoid comparing apples and oranges.

**Additional Comments On Reviewer Discussion:**

There was not rebuttal and no discussion between the reviewers and the authors.

---

### Decision · Program_Chairs · 2025-01-22

Reject